# Biomechanical Characterization of Preparation for Airs above the Ground: A Mixed Approach

**DOI:** 10.3390/ani14020189

**Published:** 2024-01-06

**Authors:** Sophie Biau, Marine Leblanc, Eléna Pycik, Benoît Pasquiet, Benoit Huet

**Affiliations:** 1French Horse and Riding Institute, Saumur Technical Platform, F-49400 Saumur, France; elena.pycik@ifce.fr (E.P.); benoit.pasquiet@ifce.fr (B.P.); 2Movement—Interactions—Performance, MIP, UR 4334, Nantes Université, F-44000 Nantes, France; marine.leblanc@univ-nantes.fr (M.L.); benoit.huet@univ-nantes.fr (B.H.)

**Keywords:** phenomenology, sensorimotor empathy, air above the ground, contact, balance, hoof-beat

## Abstract

**Simple Summary:**

Equitation in the French tradition is a school of riding that emphasizes harmonious relations between humans and horses. The best-known community is the Cadre Noir of Saumur, whose specialty is the air above the ground (AAG). For the quality of the performance and transmission to future *écuyers* (i.e., elite riders of the Cadre Noir of Saumur, France), the definition of indicators allowing the biomechanical characterization of the preparation phase of (AAG) emerges as decisive. In this study, three classes of indicators were detected based on the empirical perception of the *écuyers* and quantified from curb and snaffle rein tensions, horse trunk movements, and acceleration of the four limbs. The three classes of indicators, contact, balance, and hoof-beat, were characterized during the three seconds preparation phase before forty-nine AAGs, performed by five horses worked by two *écuyers*. These three classes of indicators made it possible to identify a pattern of *écuyer*–horse interaction. Their action was characterized by a peak on the inside curb rein. They considered that their horse was in balance. For AAGs perceived as satisfactory by the *écuyer*, during the three seconds that followed until the movement began, the horse’s trunk was stable and the *écuyers* released the contact. Comparing the perception of the *écuyers* expressed by a method of self-confrontation interviews with biomechanical measures objectified the expert *écuyers’* feeling of developing aptitudes to adjust the human–horse interactions for improved transmission to young *écuyers*.

**Abstract:**

Equitation in the French tradition is a school of riding that emphasizes harmonious relations between humans and horses. The best-known community is the Cadre Noir of Saumur, whose specialty is the air above the ground (AAG). No study has yet looked at the horse–rider interaction in this specific context. The purpose of this study was to identify and quantify indicators of AAGs based on the empirical perception of the *écuyers* expressed by a method of self-confrontation interviews. Fourteen training sessions were the subject of phenomenological and biomechanical approaches. Contact, balance, and hoof-beat, decisive for performance quality, were characterized for 49 AAGs, performed by five horses trained by two expert *écuyers*, with rein tension meters integrated in their double bridle (curb and snaffle reins) and six inertial measurement units fixed on the limbs, sternum, and croup. Their action was characterized by a peak of 65 ± 39 N on the inside curb rein. They considered that their horse was in balance (forehand inclined 13 ± 7° and −12 ± 9° for the hind hand). After the peak, during the 3.3 ± 2 s the horse’s trunk was stable and the *écuyers* released the contact until the AAG was perceived as satisfactory by the *écuyer*. The mixed approach allowed a pattern of action to be envisaged for the *écuyer* based on contact, balance, and hoof-beat in the execution of AAGs. The quantification of rein tension, trunk movements, and acceleration of the four limbs objectified the expert *écuyers’* feeling of developing aptitudes for their actions in the human–horse interactions for improved transmission to young *écuyers*. The mixed approach used in this study has given rise to new training methods that are transferable to other equestrian activities.

## 1. Introduction

Inscribed in 2011 on the Representative List of the Intangible Cultural Heritage of Humanity [1], equitation in the French tradition is a school of riding that emphasizes harmonious relations between humans and horses. Knowledge of the animal itself (physiology, psychology, anatomy) and human nature (emotions and the body) are complemented by a rider’s state of mind that combines skill and respect for the horse. The most widely known community is the Cadre Noir of Saumur, based at the Institut Français du Cheval et de l’Équitation, in Saumur, France, one of whose specialties is the air above the ground (AAG).

AAGs are performed in order to reinforce the *écuyers*’ seat and bravery. For the in-hand conditions, riders are also trained in AAGs to develop their dexterity and tact and to preserve equitation in the French tradition. The three AAGs are the croupade, the courbette, and the capriole [2,3]. F. Robichon de La Guérinière (1688–1751) was the first to describe them [4]. In the croupade, the horse, in the hand, is mobilized from a piaffe with a rather low forehand, and energetically throws its back legs out behind. In the courbette, the horse, in the hand, is mobilized from a piaffe with a rather high forehand, and goes up on its hind legs, elevating its forehand. Finally, in the capriole, from the *terre-à-terre* canter, the horse first raises its forehand and then bounds forward and vigorously throws out its hind legs behind [3]. At Saumur’s Cadre Noir, the AAGs are executed in both ridden and in-hand conditions.

Contact is defined as a connection between hands of the rider and the mouth of the horse. Unlike in the ridden condition, this in-hand condition is characterized by a substantial reduction in contact with the animal, limited to the mouth through the reins and to the touch points with the whip. The capacity to establish a “good” contact is built through a long appropriation by the *écuyer* of how the horse reacts and how to take this into account to act appropriately [5]. This progressive construction so as to “feel the horse” through the contact, especially in the preparation phase, is an important challenge in the *écuyers*’ in-hand training. The possibility of relying on biomechanical indicators during the preparation phase would represent a definite contribution to the understanding of the favorable conditions for performing AAGs well. In order to test training methods, some studies have focused on in-hand tensions [6,7,8]. But to our knowledge, no study has looked at the horse–rider interaction in the specific context of airs above the ground, which impose specific constraints on the horse–*écuyer* dyad. Moreover, only one study has examined the biomechanical characterization of AAGs [9]. Based on a kinematic analysis (system of motion capture, 100 Hz) with seven horses, results showed specific and common characteristics. The characteristics of engagement and explosive power of the movement of these specific movements require years of training for the young *écuyer* and their horse to achieve in respect of equitation in the French tradition.

The aim of this present study was to contribute to the definition of indicators allowing the biomechanical characterization of the preparation phase of AAGs, decisive for performance quality.

The first step was to identify indicators based on the empirical perception of the *écuyers*. We can expect indicators linked to the components of the French doctrine described by General L’Hotte, chief *écuyer* at Saumur from 1864 to 1870 [10]. He described the forces that produce impulsion, the balance and position that encourage the slowing or stopping movement, and a good connection with the rider accompanying the horse through all the movements. In this study, hoof-beat (which could illustrate the forces involved), hip engagement (which could illustrate the balance and position), and hand action, i.e., the contact, (which could illustrate the connection), should therefore be perceived by the *écuyers*. These components have already been measured in the literature with biomechanical methods in order to evaluate horse–rider interaction. These methods comprise kinetic measurements with force plates or kinematic measurements using motion analysis, accelerometers, or inertial measurement units [11]. The contact has been extensively studied by measuring rein tension [6,7,12].

The second step was to quantify these indicators in the context of their practice by measuring rein tensions, the horse’s position, and locomotion, and to describe factors of interaction in this *écuyer*–horse system such as the characteristics of the *écuyer* or the type and quality of exercise [6,7].

Cooperation between generations of *écuyers* is strong, with respect for the experience of the oldest. Education/transmission is one of their missions, and new knowledge enriches the UNESCO register of the Intangible Cultural Heritage of Humanity. The challenge was to objectify what the *écuyers* express, and to identify patterns in their actions. The aim was to calibrate perceptual experiences described in syncretic, metaphorical terms and to objectify dimensions of the activity that are not perceived as significant by the *écuyers* and that do not contribute to their pre-reflexive consciousness while participating in the horse–rider interaction. New training methods may emerge from this and be transferable to other equestrian activities.

## 2. Materials and Methods

### 2.1. Background to the Development of Methods

Between September 2020 and February 2022, over 100 regular training sessions were recorded and analyzed. Retrospective verbalization data were collected in self-confrontation interviews. Some of them were enriched with measurements synchronized with the session videos following each training session. The analysis of these interviews allowed for the identification of three categories of indicators: contact, balance, and hoof-beat. These categories of indicators were referenced from a prior study [5] and emerged from the verbalizations of the *écuyers* during thirty interviews. All interviews were selected because they delved into the notion of contact, which is central to equestrian technical culture. The concept of contact proved to be significant in the *écuyers*’ experiences. Therefore, based on these enriched self-confrontation interviews described in [13], a mind map was created to explore the meanings attributed by the *écuyers* to the notion of contact and to identify their perceptions through contact, which characterize optimal experiences in their interaction with the horses. These mind maps revealed that the perception of contact, conveyed through sensitivity to rein tensions, is intrinsically linked to the perception of balance, impulsion, or foot beat. These three indicators play a crucial role in establishing a “good” contact. The present study was based on phenomenological data constructed from enriched self-confrontation interviews. The *écuyers* spontaneously began to describe the interaction using the three indicators a few seconds before the AAG. The authors termed this sequence the “preparation phase”. This process was carried out prior to the study and led to identification of indicators based on the empirical perception of *écuyers*.

### 2.2. Subjects and Procedure

Of the 100 regular training sessions that were the subject of the first methodology, only 14 of them, chosen at random, were the subject of biomechanical measurements with transducers in order to assess contact, balance, and hoof-beat, the three classes of indicators designated by the collection of retrospective verbalization data. The sessions lasted 14 ± 5 min (excluding the warm-up) during which the horses performed 9 ± 5 AAGs. During the warm-up, horses were lunged or ridden by their *écuyer* and did not perform any AAGs. The study was based on an analysis of 49 AAGs, performed by five horses worked by two expert *écuyers*. The two expert *écuyers* were male. At the time of this study, the first *écuyer* (*Écuyer* G) was 52 years old and joined the Cadre Noir in 1998. The second *écuyer (Écuyer* V), aged 47, joined the Cadre Noir in 2001. The horses (10 ± 4 years) in the study were trained and worked by their *écuyer*, using the same equipment and the same working methods. For this study, the approach is systemic with the work of a *écuyer*–horse pair in their usual working environment. The measures took place at the Cadre Noir area where the *écuyers* habitually train. The horses were equipped with their custom saddle and reins. The *écuyers* provided written consent and the horses were deemed sound (<1 of 5 on the AAEP scale) by their veterinarian, who monitored their health throughout the study.

### 2.3. Data Collection

#### 2.3.1. Motion Transducers

Each horse was equipped with six inertial measurement units (IMU) (Shimmer3 IMU Unit, Shimmer Sensing, Dublin, Ireland) (Figure 1). The Shimmer3 IMUs were composed of several 3D accelerometers, a 3D gyroscope and a 3D magnetometer, and an integrated motion processor for onboard 3D orientation estimation (Madgwick algorithm), and sampling at 256 Hz. One IMU was glued on the croup with tape. The position, identified visually, corresponds to a part of the anatomy with a rather flat surface. This part was located by touch. Another IMU was attached to the girth, against the sternum. The visual verification of IMU locations in the middle, on both sides of the tail, and on the ventral median line was carried out before any recording. As the IMUs are considered to be joined to the anatomical part on which they are fixed, analysis of the IMU on the croup can be considered as the analysis of the hindquarters’ movement, and that of the IMU on the sternum as the forehands’ movement. The four other IMUs were placed on the two metacarpal and two metatarsal bones. They were fixed by an elastic holster attachment around the boots, bound by tape. The four IMUs of the limbs can weigh up to 200 g, and those of the trunk up to 16 g.

#### 2.3.2. Rein Tension Transducers

A rein tension meter was used to collect rein tension data. It consisted of a transducer and a Shimmer acquisition box (Shimmer3 Bridge Amplifier+ Unit, Shimmer Sensing, Dublin, Ireland). The whole device was integrated to the reins. The transducers (developed by the GeM UMR CNRS 6183—Nantes Université—Centrale Nantes—CNRS Laboratory in France https://gem.ec-nantes.fr/, accessed on 1 February 2020) were based on a double curvature principle made of composite materials, equipped with 350 Ohms strain gauges, and wired in full bridge. The sensors had a measurement range of 1500 newtons and a sensitivity of 5 mV/V full scale. The values were recorded by Shimmer acquisition boxes (amplifier gain: 183.7 ± 1%).

The four reins, I.e., the left curb rein (LCR) corresponding to the inside curb rein, right curb rein (RCR), left snaffle rein (LSR), and right snaffle rein (RSR), were equipped with this meter (Figure 1). For safety reasons, the reins were not cut to install the strain gauges so that if the latter broke, the reins would remain intact. The meter was calibrated before each data collection session by suspending known weights ranging from 0 to 150 N. The raw data were sampled at 256 Hz and stored on the microSD card in the acquisition boxes. The six IMUs and the four acquisition boxes were synchronized (master/slave synchronization). The maximum synchronization error was 10–20 ms. The tensions within each rein, expressed in newtons (N), were recorded continuously during testing. Each of the reins equipped with the sensor weighed under 200 g. Any forces that measured less than or were equal to 200 g were considered to be zero.

### 2.4. Data Analysis

Shimmer IMU data provide an estimate of the sensor orientation as a unit quaternion. This orientation is computed by sensors themselves from accelerometric data to ±16 g, gyroscopic data to ±2000 deg/s, and magnetic data to ±49.152 gauss using the Magdwick algorithm [14]. This orientation is the rotation from a global reference frame to a local reference frame (IMU reference frame). From this orientation, we compute the up–down inclination of the horse’s forehand (sternum) and hind hand (croup), as described in Appendix A. We define up–down inclination as the angle between the rearward horizontal direction and the IMU *x*-axis projection in a vertical plane. In this study, for the sternum, a positive inclination angle can be equated with a nose-down orientation and a negative inclination angle can be equated with a nose-up orientation of the forehand. For the croup, a negative inclination angle can be equated with a croup-down orientation. The more negative the inclination angle, the more the hind hand is engaged.

For the limb IMUs, only the acceleration measured directly in the IMU reference frame is considered. The acceleration vector is proportional to the resultant force applied to the body where the sensor is attached [15]. Only the value of the last maximum peak before the AAG was used.

All data were analyzed with the Matlab R2019b software (MathWorks^®^, Portola Valley, CA, USA).

#### 2.4.1. Delimitation of the Preparation Phase 

The AAGs of the 14 sessions recorded were cut out. Signals of accelerations and forces were cut off five seconds before the beginning of each courbette and croupade. This time interval was decided upon based on the verbalization of the *écuyers*. Moreover, the use of the enhanced videos during this first methodological step also made it possible to identify a peak in LCR tension. The preparation phase was delimited as follows: the beginning of the preparation phase corresponds to the end of the peak in LCR tension, and the end of the phase corresponds to the beginning of the AAG, marked by an abrupt change in the angle of inclination of the forehand (Figure 2).

#### 2.4.2. Calculated Variables

Calculated variables (Table 1) were chosen based on those revealed by *écuyers* on mind maps. These indicators in relation to contact, balance, and hoof-beat were interpreted as biomechanical parameters commonly used in the literature.

In addition to the acquisition of data from the IMUs and force sensors, all the sessions were video recorded using standard equipment. These videos were used as an aid for the debriefing with the two *écuyers*. They described each courbette and croupade as satisfactory (+) or unsatisfactory (−). The subjective evaluations of the *écuyers* regarding the AAG were based on various criteria: the contact, the rhythm, the balance, the height, and fluidity of the horse’s movement.

### 2.5. Statistical Analysis

Mean values and standard deviations were calculated for 49 AAGs, for the 29 courbettes, 20 croupades, 15 satisfactory courbettes, 14 unsatisfactory courbettes, 10 satisfactory croupades, 10 unsatisfactory croupades, 30 AAGs performed by *Écuyer* G, and 19 AAGs performed by *Écuyer* V.

The statistics were processed using Xlstat (Addinsoft^®^.2022.1.2). The data were assessed for normal data distribution via the Shapiro–Wilk test. Only seven variables follow the normal distribution (LCR_mean, LCR_std, FE_croup, FE_croup_mean, F_peaks, and H_peaks). Non-parametric tests were therefore chosen to describe the *écuyer*–horse interaction and to evaluate the “*écuyer*” factor, the “type of exercise” factor, and the “quality of AAGs” factor (satisfactory vs. unsatisfactory, expressed by the two *écuyers* during their interviews), more precisely the Mann–Whitney U test (*p* < 0.05) and the Spearman’s test for correlations between variables.

## 3. Results

The statistical results are shown in Table 2. None of the factors studied (*écuyer*, quality of AAG, type of AAG) had a significant impact on the duration of the preparation phase (PP_Time = 3.3 ± 2 s). The same was true for the LCR_peak, LCR_mean, RCR_std, FE_sternum, and FE_sternum_mean. The tensions were inversely correlated with the inclination movement of the forehand: the more the forehand was nose-down inclined, the lower the RCR tensions and their variations (respectively, −0.301, *p*-value = 0.047; −0.413, *p*-value = 0.006). In addition, for the croupade, RSR tension was correlated with F_peaks (−0.818, *p*-value = 0.034), FE_Croup_mean with LCR_mean (0.926, *p*-value = 0.012), and LCR_std (0.873, *p*-value = 0.024). The duration of preparation (PP_Time) was correlated with engagement (FE_croup_mean) (0.821, *p*-value = 0.048). After the 65 ± 39 N LCR peak, the contact was less than one kg until the AAG: LCR_mean = 6 ± 4 N, RCR_mean = 4 ± 4 N, RSR_mean = 6 ± 6 N, and LSR < 0.5 N. The forehand attitude at the beginning of the preparation phase (FE_sternum) was 13 ± 7° and remained stable throughout (FE_sternum_mean = 13 ± 9°).

### 3.1. Courbette vs. Croupade

The preparation of the courbette differed from that of the croupade by the higher hind limb accelerations (H_peaks = 237 ± 107 vs. 111 ± 106 m/s^2^, *p* = 0.001) as well as more sustained contact with the RSR (RSR_mean = 8 ± 7 vs. 3 ± 4 N, *p* = 0.012) and its variations (RSR_std = 5 ± 4 vs. 2 ± 2 N, *p* = 0.010).

### 3.2. Écuyer V vs. Écuyer G

The two *écuyers* differed by the right-side tensions, balance, and hoof-beat.

The *Écuyer* V used the right curb rein (RCR_mean: 5 ± 4 vs. 3 ± 3 N, *p* = 0.017) more compared to *Écuyer* G. During the interview, *Écuyer* V said: “Keep the contact on the outside rein so that the horse moves forward; if the rein isn’t taut enough, the horse isn’t engaged enough and the AAG is twisted; if there isn’t enough tension on the outside rein, then the horse isn’t straight.”

*Écuyer* G used the right snaffle rein more (RSR: 3 ± 4 vs. 8 ± 6 N, *p* = 0.001).

The inclination of the croup (FE_croup: −16 ± 9 vs. −10 ± 8°, *p* = 0.018) at the time of the peak LCR tension and during the 3 s that follow until the AAG was more pronounced with *Écuyer* V.

*Écuyer* G had horses whose forehand (F_peaks: 276 ± 149 vs. 225 ± 80 m/s^2^) and hindfoot (H_peaks: 247 ± 94 vs. 88 ± 97 m/s^2^) hoof-beats were significantly higher than those of *Écuyer* V’s horses.

### 3.3. Quality of the AAGs (Unsatisfactory vs. Satisfactory)

The preparation phase of unsatisfactory croupades was characterized by a greater use of the LCR (LCR_std: 9 ± 5 N vs. 4 ± 3, *p* = 0.004) (Figure 3) and a higher average tension of the RSR (RSR_mean: 5 ± 5 vs. 1 ± 1 N, *p* = 0.017). Unsatisfactory courbettes were characterized by a greater variation in the RSR (RSR_std: 7 ± 4 vs. 3 ± 3 N, *p* = 0.022).

## 4. Discussion

### 4.1. The Three Classes of Biomechanical Indicators of the Pattern of Écuyer–Horse Interaction in the Performance of AAGs

The results of this study were obtained from two types of analysis, sensorimotor and biomechanical. They are interdependent, since the first method, consisting in a methodological process to explore the sensorimotor dimensions, made it possible to identify three classes of indicators of AAG performance. It was on these criteria that the biomechanical analysis was built. Indeed, the first step was aimed at exploring the sensorimotor dimensions of the interaction between humans and horses with their environment. It was assumed that rider–horse interaction is achieved through contact, the connection between the rider’s hands and the horse’s mouth, and the rider’s position, legs, and seat. Cadre Noir *écuyers* work on AAGs mainly in hand to respond exclusively to contact, with the whip being used essentially as a cue to trigger the AAGs. As with working on long reins, this involves depending on the reins as the main form of communication. Direct contact then comes down to the hand–mouth relationship, which becomes predominant in the connection with the horse, and in establishing the attitude required to execute the AAGs. As with long reining, one might expect higher tension values than those obtained when the horse is ridden [7]. The quality of the connection with the horse depends on the *écuyer*’s ability to perceive the horse’s activity, to “feel” it and to act accordingly [5]. 

These dispositions were studied from a phenomenological perspective through interviews aimed at elucidating the subtle sensorimotor dimensions at play in their interactions with the horses. The questioning used in these interviews, for instance, aimed to elicit the nuanced sensations perceived by the *écuyers* through contact, at a specific moment during the session, experienced as particularly significant by the actors (e.g., before or during a movement perceived as satisfying by the *écuyer*). Thanks to this phenomenological perspective, it was shown that expert *écuyers* have each developed a sensorimotor empathy [5], which is a disposition to perceive and understand the horse through their own bodies and to adjust with it finely. It is this sensorimotor empathy that allows the *écuyer* and the horse to connect and execute complex performances.

In addition to contact, the *écuyer*’s disposition to perceive their horse also highlighted the horse’s attitude (balance) and the concept of rhythm (hoof-beat).

This notion of rhythm expressed by the *écuyers* probably has to do with the fact that managing the distribution of vertical forces between the four limbs is one of the fundamental motor control strategies for mastering pitching in the sagittal plane [16,17]. Strategies to keep balance include adjusting the timing of limb contact, hoof placement, and modifying the distribution of the ground reaction force between the fore and hind limbs. Pitching during the preparation phase of the courbette, collection (FE_croup), and the hindlimb footbeat (H_peaks) was necessary to raise the withers to a speed of 1.3 m/s, the speed calculated for a reference courbette [9].

As far as the concept of balance, the attitude sought by the two *écuyers*, whatever the AAG, corresponded to a sternum inclination of 13 ± 7° and a croup inclination of −12 ± 9°. In an interview with one of the *écuyers*, he described this attitude using the following expression: “he [the horse] has a round back”. The same *écuyer* also differed from the other *écuyer* by the greater engagement of the hindquarters (FE_croup_mean). The *écuyers* worked their horses during the mobilization phase (the piaffe was not analyzed in this study) to achieve the optimal attitude and thus promote the explosiveness expected from the AAG. When the *écuyer* judged the horse’s attitude to be satisfactory, he applied a pull of 65 ± 39 N to the LCR and then three seconds elapsed before the AAG was executed. During these three seconds, the tensions of the four reins were very low (RCR_mean = 4 ± 4 N; LCR_mean = 6 ± 4 N; RSR_mean = 6 ± 6 N; LSR_mean < 0.5 N), which for the *écuyers* corresponds to a light contact. The horse kept its balance. The trunk was relatively stable (FE_croup_mean = −15 ± 6° and FE_sternum_mean = 13 ± 9°). This stability may well help to maintain the orientation of the vestibular apparatus in relation to gravity [18], to better orient the trunk moments in the direction of movement.

The three classes of biomechanical indicators of the pattern of *écuyer*–horse interaction in the performance of AAGs correspond to the components of the French doctrine: the forces, perceived by the *écuyer* with the notion of hoof-beat; the balance, perceived with the position of the forehand and haunches as well as the pitch; a good connection between the *écuyer* and the horse perceived by the contact and measured by the tension of the reins. Following the results of this study, and in particular the showing of the 65 N peak of tension, we could envisage a joint effort by rider and horse to reduce the value of this peak to improve the horse’s comfort [19].

### 4.2. The Role of the Reins

The general consensus of previous studies was that rein tension increased with the gait of the horse. The highest rein tension occurred at the canter, followed by the trot, with the lowest in the walk. Compared to the ridden test, the values were lower for the unridden test [6] and higher for the long-reined test, for some figures [7]. Although values from other studies should be compared with caution due to the method of calculation, the difference in equipment, the level of horses, the exercise [12], the mean value of the contact in this study during the AAG preparation phase corresponded to a tension equivalent to that of an unridden walk (6.5 N ± 2.8 N [6]) or a ridden walk (6.9 to 43 N [12]). The expressions used by the *écuyers*, such as “to move forward with the outside rein” and “to balance with the inside rein” provided information on the role of the reins. The tensions of the RSR may well predominate during the mobilization phase of the piaffe or of the terre-à-terre to “make the horse go forward” [10], not considered in this study. During the preparation phase, the right-hand snaffle tensions were low (6 ± 6 N), as the horse was not moving. It was the left curb rein that took over with a peak of 65 N. Once the horse had reached the required attitude and immediately after this peak in tension of the LCR, the *écuyers* released the contact. If the balance was incorrect, the *écuyers* used the reins again during the three seconds, but finally the AAG was considered unsatisfactory. The expressions used by the *écuyers*, “to move forward with the outside rein” and “to balance with the inside rein” then take on their full meaning. The value of the peak of the LCR of 65 N, similar for both *écuyers* and for all the AAGs, was used to determine the start of the preparation phase of the AAGs. This peak seems to be used by the *écuyers* to “provide an overview” and at the same time probably to inform the horse that the AAG is imminent. The tension of the right snaffle rein was greater for the courbette, which in this case can be explained by the fact that the horse pulls during the preparation of the forehand raise.

On the other hand, it was the action on the left curb rein that was used more for the croupade. This tension can be explained by the *écuyers*’ action on the rein, used to signal to the horse to lower the forehand to achieve the appropriate balance that will allow it to release the croupade. The measurement system used to measure the tensions did not confirm the action of the *écuyer* nor that of the horse. Furthermore, the RSR and RCR tension values are only indicative and reflect the tension in the bit and not the action of the *écuyer*. With the *écuyer* close to the left side of the horse, the reins rub more or less on the neck depending on the attitude of the forehand. In this study, the correlations between forehand inclination and RCR tension could be explained by this friction. The position of the neck modified the force of friction on it. The more the forehand was nose-down inclined, the less the rein rubbed against the neck and the tension decreased. The left snaffle rein was probably not used. The *écuyer*’s hand was close to the horse’s mouth and the tension values on the left snaffle rein were too low to distinguish a contact of less than the weight of the sensor plus rein.

### 4.3. Regularity and Variety in the Work of Expert Écuyers

Quantification of the variables enabled the identification of a pattern of interaction between *écuyer* and horse in the execution of AAGs by these two experts. The biomechanical measurements made it possible to characterize the dimensions of the interaction between the *écuyer* and the horse (e.g., contact by measuring the tension of the four reins) or the attitude and movements of the horse resulting from this interaction (e.g., a lowering of the croup, or installation of a rhythm) which, in the preparation or execution of the AAGs, were found regardless of the *écuyer* or the AAG, and thus reveal invariances in the preparation of AAGs.

Nevertheless, the measurements also highlighted differences in the attitude of the horses of *écuyers* V and G (FE_croup, FE_croup_mean) and hoof-beat (F_peaks, H_peaks) which reveal the singularity of the *écuyers*’ practice and the existence of preferences in how to establish interaction with the horse, considering such or such aspect of its attitude and acting to constrain its activity. For example, *Écuyer* V seemed to be more demanding regarding the engagement of the croup, whereas *Écuyer* G mobilized the forehand with his RSR more during the three seconds of the preparation phase and solicited more fore and hindlimb activity. It was assumed that the propulsion of the hindlimbs increases the pitching of the forehand [16]. Biomechanical measurements are therefore twofold in interest: (a) they make it possible to identify what is common to the *écuyers* in this phase of preparation for the AAGs, which is crucial for the quality of the latter, and (b) they make it possible to distinguish between the regularities of the practice and the singularities of the style [20] specific to each *écuyer*.

### 4.4. The Difficult Biomechanical Characterization of AAG Quality

The test of the effect of the quality of each AAG (Table 2) highlights the importance of the RSR and the LCR during the preparation phase. This contact, which increased for unsatisfactory AAGs, may reflect an action by the *écuyer* to adjust, in addition to the impulsion, the balance and/or a reaction by the horse “pulling” (a distinction that cannot be verified by the sensors used). When contact increased during the preparation phase, the AAG was generally considered unsatisfactory by the *écuyers*. In this study, few variables distinguished between AAGs considered satisfactory and AAGs considered unsatisfactory by the *écuyers*. The interviews revealed a great diversity of judging elements, which referred either to the preparation (e.g., the mention of a “stolen courbette” when the horse performs the AAG before the *écuyer*’s signal, or of a “light contact” during the preparation phase of the AAG), or to the AAG itself (e.g., the mention of a “not high enough courbette”, or the notion that the horse “comes back down well in its tracks”).

The assessment made by the *écuyers* also considered other elements, such as the level of training of the horse (i.e., a certain AAG may be judged satisfactory for a young horse whereas it would have been judged unsatisfactory for an already trained and experienced horse), or the horse’s state of fitness. These elements, taken into account in the subjective assessment of the quality of an AAG, made it extremely difficult, in terms of biomechanical analysis, to compare satisfactory or unsatisfactory AAGs. Indeed, there is no standardized nomenclature enabling a “satisfactory” or “unsatisfactory” AAG to be characterized based on biomechanical variables, unlike an assessment carried out by a judge, for example [21,22]. The variables measured did not make it possible to identify the factors influencing the assessment made by the *écuyers*. The classification of satisfactory and unsatisfactory AAGs in this study is therefore open to criticism.

### 4.5. The Interest of Mixed Methods for Understanding Human–Horse Interactions

All things considered, measurement has enabled us to calibrate the perceptions of *écuyers*, for example in terms of rein tension, to understand the meaning of the words they use, and sometimes to qualify them. For example, a “light” contact for the croupade is not exactly the same as for the courbette from the viewpoint of the mechanical characterization of rein tensions. *Écuyers* “accept” more tension in the RSR for the preparation of the courbette than for the croupade. In this case, the fact that there is more tension does not mean that the contact is not as good. These elements clarify and lend perspective to the notion of “lightness” of contact [23]. They lead to the recommendation of a radically situated approach to contact, taking into consideration the respective experiences of the *écuyer* and the horse, and the quality of their connection, the product of the history of their relationship, particularly in work.

Contact during the preparation phase is a determining factor in obtaining a satisfactory AAG, particularly contact with the RSR for the courbette and the LCR for the croupade, and appeared as such in the phenomenological data, which highlighted the *écuyers*’ preoccupations and focus during this phase. The measurements also revealed the perceptual-motor finesse of the *écuyers* through contact with, for example, a peak LCR of 65 N for both of them and for both types of AAG. This value of contact with the LCR is useful objective information for passing on know-how to the student *écuyers*. The cross-referencing of biomechanical measurements with phenomenological data from the analysis of experience, already used in the domain of sports science [15,16,17,18,24,25,26], has thus proved to be of great value in understanding *écuyer*–horse interactions in AAGs. Beyond this specific context of practice, crossing methods emerges as a promising way of studying human–horse interactions in riding practices, in that it allows us to better account for their complexity through the reciprocal enrichment of analyses. Based on the results of this study, the *écuyers* are considering new training methods to enhance the training of young *écuyers*. The creation of capsules with videos enriched with biomechanical data, such as rein tensions, is being envisaged. Additionally, they are considering the implementation of cross self-confrontation interviews [27] aimed at confronting the experiences of the trainer *écuyer* with those of the trainee *écuyer*. These training methods are transferable to other equestrian activities.

### 4.6. Limitations

The lack of homogeneity in the numbers involved was a bias in the statistical analysis. The results of the Mann–Whitney tests should be considered as possible interpretations. The RSR values that were greater for one of the *écuyers* can be explained by the fact that this *écuyer* performed more courbettes (22) than croupades (8). The same applies to the hindlimb beat, which was higher for this *écuyer*’s horses. The RSR tension values and the hindlimb acceleration peaks were two variables that differed significantly between the two AAGs. The diversity in the number of AAGs performed by the five horses meant that it was not possible to test the effect of the horse on the variables calculated. However, the tension values measured do not indicate whether it is a hand action and/or a horse action. We might have expected higher tension values for certain horses [19], which may perhaps explain the standard deviation of 39 N for the peak value of the LCR of 65 N at the beginning of the preparation phase. In addition, the direction of the pull is not given (downwards, upwards, in the direction of movement or not). This information could have completed the *écuyer*’s empirical analysis.

Finally, this pattern must be considered incomplete. It did not reflect all the data collected by the *écuyers*. Expressions such as “make the horse adopt my speed” highlighted the importance of the *écuyer*’s body, attitude, and movements. In addition, objective assessments of the horse’s comfort and discomfort during this type of work would complete the multidimensional approach. Further measures need to be considered to increase our knowledge of the interaction between the *écuyer* and the horse in the performance of AAGs so that this skill can be better transmitted.

## 5. Conclusions

The methodological originality of this study consisted in mixing an approach of activity analysis by self-confrontation interviews and a biomechanical approach. This multidisciplinary study enabled an objective description of a pattern of horse–*écuyer* interaction during the preparation of the AAGs. The *écuyers* mobilized their horses to achieve optimal balance. This balance was checked by a peak of tension on the left curb rein. After this peak, the tension on the four reins was released and the horse maintained its balance by moving its limbs. The horse performed the AAG three seconds after. If these conditions were not respected, the *écuyer* judged the AAG unsatisfactory. More broadly, understanding how *écuyer*–horse synergies are established enables a better grasp of how to study human–horse synergies. Indeed, the presence or absence of the favorable pattern in the preparation phase, identifiable through variations in rein tensions, emerges as an operational reduction. This reduction serves, on the one hand, to explore the complexity of a multimodal contact (i.e., perceived rein tensions, balance, impulsion, cadence) and, on the other hand, to assess the quality of the contact. This study provides elements to enrich the training of future *écuyers* and more generally to develop aptitudes to act in the interactions of human and horse.

## Figures and Tables

**Figure 1 animals-14-00189-f001:**
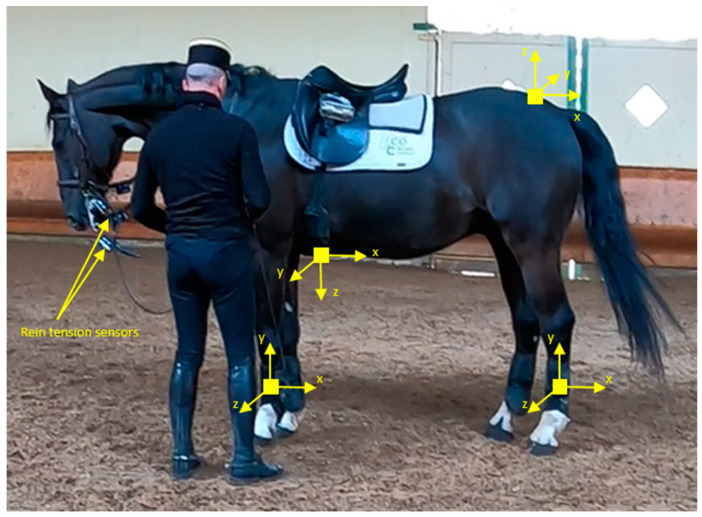
The horse is equipped with six IMUs (one on each limb, one fixed against the sternum, one glued on the croup) and four rein sensors (left and right curb reins and snaffle reins).

**Figure 2 animals-14-00189-f002:**
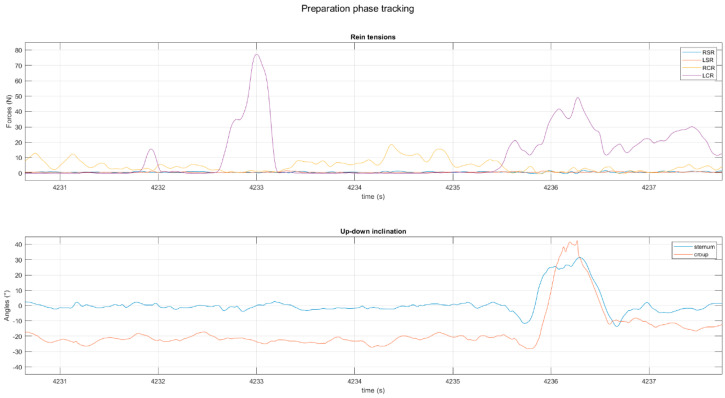
(**Top**) tensions of left curb rein (LCR), right curb rein (RCR), left snaffle rein (LSR), right snaffle rein (RSR), and (**bottom**) sternum and croup inclination angles. For the sternum, a positive inclination angle can be equated with a nose-down orientation and a negative inclination angle can be equated with a nose-up orientation of the forehand. For the croup, a negative inclination angle can be equated with a croup-down orientation. The two blue dotted lines mark the preparation phase (from the end of the peak LCR tension to the beginning of the rotation of the forehand) of a croupade.

**Figure 3 animals-14-00189-f003:**
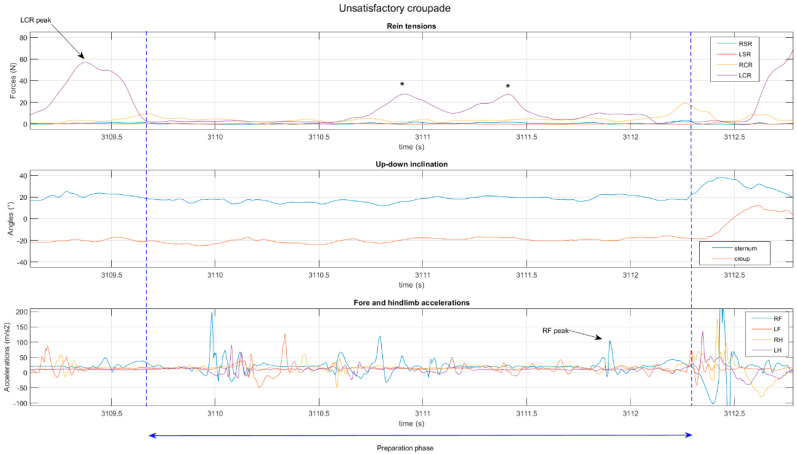
Contact (**top**), balance (**center**), and hoof-beat (**bottom**) variables of a croupade judged to be unsatisfactory by the écuyer. He described a lack of hoof-beat, particularly of the hindlimbs, which he tried to re-establish with the LCR (*).

**Table 1 animals-14-00189-t001:** List of calculated biomechanical variables and their description.

Variables	Description
PP_Time (s)	Preparation phase duration, expressed in second
**Contact**	
LCR_peak (N)	Maximal value of the left curb rein tensions for the five-second period before the start of the AAG, expressed in newtons
LCR_mean (N)	Mean value of the left curb rein tensions during the preparation phase, expressed in newtons
LCR_std (N)	Standard deviation of the left curb rein tensions during the preparation phase, expressed in newtons
RCR_mean (N)	Mean value of the right curb rein tensions during the preparation phase, expressed in newtons
RCR_std (N)	Standard deviation of the right curb rein tensions during the preparation phase, expressed in newtons
LSR_mean (N)	Mean value of the left snaffle rein tensions during the preparation phase, expressed in newtons
LSR_std (N)	Standard deviation of the left snaffle rein tensions during the preparation phase, expressed in newtons
RSR_mean (N)	Mean value of the right snaffle rein tensions during the preparation phase, expressed in newtons
RSR_std (N)	Standard deviation of the right snaffle rein tensions during the preparation phase, expressed in newtons
**Balance**	
FE_croup (°)	Croup inclination angle at the beginning of the preparation phase, position relative to a horizontal axis, expressed in degrees
FE_sternum (°)	Sternum inclination angle at the beginning of the preparation phase, position relative to a horizontal axis expressed in degrees
FE_croup_mean (°)	Mean of croup inclination angles during the preparation phase, mean position relative to a horizontal axis expressed in degrees
FE_sternum_mean (°)	Mean of sternum inclination angles during the preparation phase, mean position relative to a horizontal axis expressed in degrees
**Hoof-beat**	
H_peaks (m/s^2^)	Vertical acceleration peak value of the right and left hindlimbs of the last propulsion before the AAG, expressed in m/s^2^
F_peaks (m/s^2^)	Vertical acceleration peak value of the right and left forelimbs of the last propulsion before the AAG, expressed in m/s^2^

**Table 2 animals-14-00189-t002:** Descriptive statistics and *p*-value of a Mann–Whitney test to evaluate differences between courbette and croupade, *Écuyer* V and *Écuyer* G, satisfactory (+) and unsatisfactory (−) AAGs, and the variables “hoof-beat” (top), “contact” (center) and “balance” (bottom). Statistically significant values are bolded.

	Statistics	Mean ± Std	Courbette	Croupade	*p*-Values Croupade vs. Courbette	Courbette −	Courbette +	*p*-Values Courbette + vs. −	Croupade −	Croupade +	*p*-Values Croupade + vs. −	*Écuyer* V	*Écuyer* G	*p*-Values *Écuyer* V vs. G
	LCR_peak (N)	65 ± 39	63 ± 42	67 ± 34	0.515	68 ± 53	58 ± 29	0.991	79 ± 31	55 ± 34	0.118	72 ± 31	60 ± 42	0.055
	PP_Time (s)	3.25 ± 1.95	2.9 ± 1.8	3.75 ± 2	0.072	3.2 ± 1.6	2.5 ± 2.1	0.134	4.3 ± 2.3	3.2 ± 1.6	0.247	4 ± 2	3 ± 2	0.692
Hoof-beat	F_peaks (m/s^2^)	317 ± 148	320 ± 137	314 ± 161	0.699	356 ± 162	286 ± 104	0.23	334 ± 161	294 ± 168	0.436	**225 ± 80**	**276 ± 149**	**0.001**
	H_peaks (m/s^2^)	185 ± 124	**237 ± 107**	**111 ± 106**	**0.001**	198 ± 91	273 ± 110	0.055	128 ± 126	86 ± 78	0.585	**88 ± 97**	**247 ± 94**	**<0.0001**
Contact	LCR_mean (N)	6 ± 4	7 ± 4.5	5 ± 3	0.249	7 ± 4	6 ± 5	0.303	7 ± 4	4 ± 2	0.077	5 ± 4	6 ± 4	0.440
	LCR_std (N)	7 ± 4	7 ± 4	7 ± 4	0.475	8 ± 4	6 ± 4	0.135	**9 ± 5**	**4 ± 3**	**0.004**	7 ± 5	7 ± 4	0.666
	RCR_mean (N)	4 ± 4	4 ± 4	3 ± 3	0.177	3 ± 3	5 ± 4	0.332	2 ± 3	3 ± 3	0.503	**5 ± 4**	**3 ± 3**	**0.017**
	RCR_std (N)	3 ± 3	3 ± 3	2 ± 2	0.176	3 ± 3	4 ± 4	0.288	1 ± 2	3 ± 2	0.184	4 ± 3	2 ± 2	0.071
	LSR_mean (N)	<0.5	<0.5	<0.5		<0.5	<0.5		<0.5	<0.5	<0.5	<0.5		
	LSR_std (N)	<0.5	<0.5	<0.5		<0.5	<0.5		<0.5	<0.5	<0.5	<0.5		
	RSR_mean (N)	6 ± 6	**8 ± 7**	**3 ± 4**	**0.012**	10 ± 7	6 ± 6	0.064	**5 ± 5**	**1 ± 1**	**0.017**	**3 ± 4**	**8 ± 6**	**0.001**
	RSR_std (N)	4 ± 4	**5 ± 4**	**2 ± 2**	**0.010**	**7 ± 4**	**3 ± 3**	**0.022**	3 ± 2	1 ± 1	0.115	**2 ± 2**	**5 ± 4**	**0.000**
Balance	FE_Croup (°)	−12 ± 9	−12 ± 10	−13 ± 7	0.482	−10 ± 12	−13 ± 7	0.804	−14 ± 9	−13 ± 6	0.81	**−16 ± 9**	**−10 ± 8**	**0.018**
	FE_sternum (°)	13 ± 7	12 ± 12	11 ± 10	0.433	17 ± 12	11 ± 12	0.229	10 ± 9	12 ± 11	0.698	11 ± 11	13 ± 1	0.423
	FE_croup_mean (°)	−15 ± 6	−14 ± 6	−16 ± 5	0.328	−14 ± 7	−15 ± 5	0.458	−16 ± 5	−16 ± 5	0.731	**−19 ± 4**	**−12 ± 5**	**<0.0001**
	FE_sternum_mean (°)	13 ± 9	12 ± 9	15 ± 9	0.328	14 ± 9	10 ± 9	0.274	14 ± 9	16 ± 10	0.567	12 ± 11	14 ± 8	0.766

## Data Availability

Data are contained within the article.

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
