# Peer review of "Biomechanical Characterization of Preparation for Airs above the Ground: A Mixed Approach"

_animals, 2024, doi:10.3390/ani14020189_

Round 1

Reviewer 1 Report

Comments and Suggestions for Authors

Line 20: Please try to improve how you convey this message. ‘Comparing the perception of the écuyers with measures objectified the écuyers’ feeling of developing aptitudes for their actions in the human/horse interactions.’

Line 23: As few scientists will know what AAG are or why horses are trained to perform them, it would be helpful if this was included in the first line of the abstract.

The abstract needs to include more details related to the method, at least based on the measurements that are then reported. Again, most scientists would not know what an inside curb rein is etc. The abstract is written very much for a knowledgeable equestrian. Please try to re-write it for a more general scientific population paying careful attention to not assume prior knowledge.

Introduction

Please start with a fuller introduction to the origin/history of these movements, why they are trained.

It is also important to expand on why, from a scientific perspective, it is useful to study the horse-human interaction whilst performing these movements, and what we can learn from them.

It would be useful to include here other studies that have investigated horse-human interactions in less highly trained horse/rider combinations performing less demanding gaits and movements. How were they quantified? What have we learned so far? This will help to build a justification or reason for performing this study beyond this being only about understanding the preparation phase for performing AAG’s.

Again, please be mindful that you do not assume prior knowledge. Many researchers will not know the Cadre noir of Saumur, France for example. They will not necessarily understand what you mean by contact with the mouth or the touch points with the whip. Images of the three AAG’s may be helpful for the reader.

Method

As both 2.1 and 2.2 appear to be part of a mixed method (and also part of the aims of this study), the methods need to be expanded upon and written such that it is clear what was done. Even if partly referred to other published studies, this information needs to be clearly stated. These studies appear to be in French and one is not published (from the references). As such, the methods for all sections need full expanding, as do the results. You can include supplementary information related to how you developed the framework and also justifications for the biomechanical measurements (related to how you compared subjective and objective measurements).

2.1: It appears from this section that you used these methods in other studies (6 and 7), but the way you have written the text including terms like ‘propose’ suggests that they have yet to be conducted? Consider re-writing this section.

For example, ‘It suggests different ways of enriching the self-confrontation interviews with videos of activity. It highlights the fruitfulness of the articulation of different modes of data construction according to an enactive approach to expert practice, in order to explore its embodied dimensions. It also opens up perspectives for the development in training of dispositions to perceive and to act within complex human-animal interactions.’

This information largely does not clearly indicate what you did, it is more of an explanation of what using multiple methods of data collection of a long time period can deliver. I assume these methods were developed from other studies besides your own, but the framework was specific to this work? Additional references from other work (besides 6-7) are needed to justify your approach.

Line 76: You use the term ‘hoof beating’ as a performance indicator, I assume as this is how it is described by the écuyers, but in most equine biomechanics literature the term that this might be compared to is foot fall timing. However, as I read through the methods it appears you have used summed peak vertical acceleration from the fore and hindlimbs. A clarification of your term to other literature would be helpful.

Are the horses and écuyers used in 2.2 the same as the ones in 2.3? More details need to be included about them. Also, you said that they were <1 lame, but when in the 100 training sessions were they assesses? At the beginning? Was this repeated? How did you establish that they continued to be sound and healthy? Did you have ethical approval for this work? Was this informed consent by the écuyers?

Although you have described the equipment, you have note described the procedures. What sis you do? Were these sessions part of the 14 from 100 sessions? If so, were these the only sessions where you collected biomechanical data or did you collect data from all of the 100 sessions?

Data analysis: Please expand on how your angles were calculated and which method you followed. It is not clear how/why you used quarternions to extract angular data relative to the global reference frame from the gyros. In addition, if you are using a global reference, then technically this is not flexion/extension, as these become inclinations (as they are relative to global coordinates and not neighbouring segments). For the sternum, this is a nose-up, nose-down inclination (of essentially the trunk), which other studies have used when describing higher level movements such as piaffe. For the pelvis, this inclination will also relate to conformational differences. How did you allow for this? You have not included any references from other methods that have used these techniques. To be confident that your data is correct you need to expand here.

Full justification for these measurements is needed, as it is not clear how the absolute angle of trunk and pelvis inclination for example have been related to balance. The variability in this measurement may be an indicator, but currently the absolute angle alone is a function of forehand elevation and pelvic position (which also includes conformation as noted previously). These would be better presented in a table.

Line 189: Based on what criteria?

The numbers of each AAG in the stats leads again to procedure. How many are performed in a training session and for how long is each session?

Line 202: significance? It is also not clear what you were trying to achieve with the correlations. The statistics should link back to your aims. How did you analyse the qualitative data? Where is the information on how your framework and descriptions of performance were established?

Did G and V écuyers train the same or different horses?

Results

The text needs to be improved using better reporting and formal language at times.  

The two line graphs are only individual examples that justify the method of data selection and show one example of the croupade judged to be unsatisfactory. As almost no scientific information is available on these movements it is important to show more graphical data than this. At the very least, mean and SD of each movement for satisfactory/unsatisfactory, but with 5 horses and 2 écuyers, there are other ways of presenting these data.

Discussion

Some of the methods that are not included in the paper are described at the start of the discussion and also what appear to be hypotheses that are not stated with the aims at the end of the introduction. This will need revising once the other sections have been revised. Here it should be stated whether your hypotheses can be accepted or rejected.

Your self citation related to the discussion of using a phenomenological method to extract sensorimotor information is insufficient. Again, until your methods are clearer it is difficult to assess how the use of questioning in your interviews has led to them developing empathy.

Line 288: It is not clear what you mean by a bent forehand. Also, these are results and link back to previous comments related to the calculation, extraction and description of these measurements.

Line 295: The mean is 65 N, but this is not the case for all data produced or indeed each écuyer. Again, these are technically results. Whilst the description of the aids applied and results movements produced by the horse provide an insight, ensure that this is a discussion rather than additional reporting of results.

Line 309: Absolute values from other studies should be compared with caution due to the difference in equipment and methods.

Line 313: Please keep the discussion focussed on your results, not on other studies. For the non-equine reader again this will be difficult to follow, so consider re-writing in order that they may follow this.

Line 325: Why on the right rein specifically?

Line 332 to 339: This appears to be an extensive description of assumptions in terms of changes in rein tension. Please support it with evidence or supporting literature.

Line 347-348: the practice of AAG preparation

Line 349: I don’t remember you including between horse differences in the results?

4.4 It would be useful to include here a comparison to other studies that have assessed quality/skill subjectively and objectively. There are several where biomechanical measures have been assessed against judged scores in dressage.

4.5 & 4.6 I can’t really comment here until additional work has been done on the paper.

Conclusion

As this study collected data from movements that have rarely been evaluated the conclusion is specific to those results, but how does this work provide valuable knowledge beyond the information presented here? It seems a wasted opportunity not to fully consider this.

Comments on the Quality of English Language

The authors need to make sure that their text is clear and easy to follow for all readers throughout.

Author Response

Dear Mr, Mrs,

Thank you very much for your interest in the paper, the time you took to review it, and the relevance of your comments.

Here are our responses:

To improve to convey the message, we have added the idea of transmission to young riders.

We've also added a presentation of the Cadre noir community and their specialty, AAGs, which was briefly presented in the summary and developed in the introduction. We have also clarified the double bridle equipment to understand curb rein.

The introduction has been expanded to better understand these movements' context, origin, and why they are still performed. The aim of the study was also detailed through the prism of man/horse interaction. Hypotheses were formulated about the principles of traditional French equitation which are based on the notions of force, balance, and contact, notions which are quantifiable and quantified in some studies. We have added references. These principles do not only concern AAGS but also other equestrian activities such as dressage. The results of the study are transferable.

To better understand what the AAGs are, we have quoted Barry J.C., a former écuyer, who describes in English, the history, traditional French riding, the Cadre Noir, etc. Readers can refer to him for further information.

About the methods, we have enriched the paragraph on the method for exploring the sensomotoric dimension by adding details and references. The écuyers defined the quality of satisfactory or unsatisfactory AAGs during the interviews.

For the biomechanical part, you will find an appendix at the end of the manuscript explaining the calculations and a more complete description of the variables in Table 1 (position at the time of the 65N peak and mean during the 3s preparation phase). The mean/std is in Table 2. This peak of 65N±39 is an average calculated for all AAGs and écuyer/horse pairs.

The horses and écuyers were the same and information about them has been added in section 2.2. Throughout the study, the horses were examined by the veterinarian at the Cadre Noir clinic, who was responsible for monitoring their health daily. In addition, the project did not include any additional sessions or special exercises. The horses were fitted with (non-invasive) sensors during 14 of the 100 sessions.  Each écuyer worked on his horse. One horse was not worked by two écuyers (except for training sessions for young écuyers, which were not included in the study). Ecuyer V's horses were compared with écuyer G's horses.

The means of duration and number of AAGs per session have been added.

The project was approved by an ethics committee (see number at the end of the document).

Regarding the term 'hoof beating', we have decided to retain the term 'hoof beat', explaining that the acceleration vector is proportional to the resultant force applied to the body to which the sensor is attached, so the term 'foot fall timing’ is not quite appropriate.

We hope that all this clarification of the introduction and the two methods used makes the discussion easier to understand.

We agree with you regarding the comparison of tension values with other studies... The review by Dumbell (2019) shows this well. A sentence has been added to express this. Furthermore, this is the reason why certain hypotheses are not supported by other studies, given the specific nature of the movements.

We hope that the clarifications in the text have contributed to a better understanding of the study.

 Sincerely

Sophie Biau

Reviewer 2 Report

Comments and Suggestions for Authors

I thought this was a great paper and it is excellent to see work of this nature being done. 

Line Comment
12 Insert (AAG) after ground
42-43 What about the 'hops'?
64 None of the other headings have a full stop
85 Did both écuyers have the same amount of experience?
111 Delete space between / and )
142 Was the LCR the inside rein?
155-171 Why was there only one peak rein tension measure?
192-194 I am not sure this is a complete sentence.
227 Is 'curve' supposed to be 'curb'?
294-295 Does this mean that the stimulus for the AAG is a 65 N pull on the LCR? Does it matter if the LCR is the inside rein or not? 
  65 N seems very high; does this vary with how the horse is going before the stimulus is applied? For instance, if the horse is sufficiently engaged, is 65 N still used or could that be decreased?
305 Delete 'And' and start the sentence with 'Compared'
323 Is 3 s imminent for a horse?
333 It is not always clear when you are referring to ridden or in-hand movements. In this line, it would seem that this part is referring to in-hand work. 
342 Does the two experts mean the two écuyers?
347 Suggest replace 'whatever' with 'regardless of'
364 Suggest replace 'These contacts' with 'This contact'
366 Could this have been measured with accelerometers on the horse's nasal plane perhaps?
415-416 Could there be a relationship between RSR tension and hindlimb acceleration?
423 Delete ', etc.'
Comments on the Quality of English Language

I only found a few grammatical errors, as listed in the line-by-line.

Author Response

Dear Mr, Mrs,

Thank you very much for your interest in the paper, the time you took to review it, and your comments.

Here are our responses:

Both écuyers are experts. We have included some informations in the text in response to your questions.
The two 'écuyers" are two expert 'ecuyer'. We have added their age and year of practice to the text. One of them is a riding master and the two 'écuyers' are responsible for training young riders.

The LCR is one of the two types of rein,  inside rein because the horseman is standing on the left side of the horse. The peak of tension on this kidney, 3 seconds before the AAG seems to be used by the écuyers to "provide an overview" and at the same time probably to inform the horse that the AAG is imminent. This peak of 65 N is not very high compared with literature values, although the measurement context is difficult to compare. With the two riders and whatever the horse, the value is identical. Now that this indicator has been measured, it is planned to work on the horses to obtain the same response with lower values.

When the horses are ready a single peak is enough and work on learning theories shows that the horse has a memory for information for around 10 seconds.

Indeed, high-frequency accelerometers on the horse's head and the écuyer's hand could provide information about the origin of the tension (the horse or the rider 'pulling'). This is a subject to be studied, for other measurement conditions as well, such as the riding horse. The limitations of tension measurement are described in this article (Clayton, 2021, https://doi.org/10.3390/ani11102875): 'Rein tension measures a pulling force applied to the rein by the horse and/or rider. Tension values alone cannot distinguish between these two sources'

The results of the correlation test showed no relationship between RSR and hindlimb acceleration but rather with accelerations of forelimbs. Hindlimb accelerations were rather correlated with LCR.

I'm sorry, I didn't quite understand your question, what about the hops?

To better understand the context we've added a presentation of the Cadre noir community and their specialty, AAGs, which was briefly presented in the summary and developed in the introduction. The introduction has been expanded to better understand these movements' context, origin, and why they are still performed. We also have enriched the paragraph on the method for exploring the sensomotoric dimension by adding details and references. For the biomechanical part, you will find you will find an appendix at the end of the manuscript, explaining the calculations and a more complete description of the variables in Table 1.

We hope that the clarifications in the text have contributed to a better understanding of the study.

 Sincerely

Sophie Biau, Phd

Reviewer 3 Report

Comments and Suggestions for Authors

I found the article interesting, but suggest edits for clarity and readability.

Please double-check all statements and properly cite sources throughout the manuscript.

Line 39-48: Please cite statements appropriately.

One paragraph introduction? Please tell reader more about AAGs and importance of studying it.

Line 61: What is the empirical perception of the écuyers? Please add text.

Line 77: Please add interviews questions/data as supplementary file.

Line 150-185: Presenting this information in form of bullets is a bit odd. Please try to present it in a different way. Maybe in a tabular form? Same for line 225-236.

Line 251-263: Please cite your discussion statements appropriately.

Line 267-276: References?

Line 279-283: Please cite.

Most of statements throughout the discussion lacking references. Please cite all of them appropriately.

Comments on the Quality of English Language

.

Author Response

Dear Mr, Mrs,

Thank you very much for your interest in the paper and the time you took to review it.

The introduction has been expanded to better understand these movements' context, origin, and why they are still performed. To better understand what the AAGs are, we have added references. We have quoted Barry J.C., a former écuyer, who describes in English, the history, traditional French riding. Scientific references relating to rider/horse interaction have also been added.

We have enriched the paragraph on the method for exploring the sensomotoric dimension by adding details and references. For the biomechanical part, you will find you will find an appendix at the end of the manuscript explaining the calculations and a more complete description of the variables in Table 1

We hope that the clarifications in the text have contributed to a better understanding of the study.

 Sincerely

Sophie Biau

Round 2

Reviewer 1 Report

Comments and Suggestions for Authors

Intro

Line 67: New paragraph ‘Contact is defined….

Line 80 to 87: Needs re-writing.

Line 130, 132, 135: Suggested, highlighted, opened (past tense for previous work).

2.1 and 2.2: I think it would be useful to clearly indicate that these parts of the project were conducted prior to this study. Maybe something like: ‘2.1 Background to the development of methods’.

2.1: I feel there is still some work to do to explain this in the context of this study, so that it is clear for the reader. I would suggest removing lines 119 to 149 completely and starting this section with with the paragraph at line 150. This paragraph is probably sufficient to explain what was done previously with your references adding in and clarification of the preparation phase.

Line 165: ….enriched self-confrontation interviews described in [13]. (or something like that).

Line 167: Please explain how the preparation phase was identified more clearly here, in a couple of sentences at the most. i.e. why the last few seconds?

2.2 is really participants and procedure (for this study), so the title again needs modifying.

Line 232 to 233: ‘The rotation from an terrestrial reference frame to an inertial reference frame’. These are not terms that are generally used in biomechanics. Usually, the terms are global coordinate system or global reference frame and local reference frame or segment coordinate system are more commonly used.

In addition, having looked up the sensors you are using, they appear to be based on typical IMU technology, using gyros, accelerometers and magnetometers that have an internal reference frame relative to whatever the IMU is attached to. This then needs to be transformed if you are interested in global coordinates using quarternions to estimate the pelvic and sternum inclinations.

Is the appendix that you have provided from your own calculations? If so, can you please explain why you are using this method to measure local coordinates when the sensors, from the technical specification, are measuring in local coordinates? If this is from reference [21] this should not be in your appendix. It just needs to be more clearly explained here.

Line 272 to 274: this is still vague, who developed the criteria? Various is vague. If the evaluations of the écuyers were their descriptions and you used their descriptions to develop these criteria, this needs explaining further. Did you analyse the descriptions using NVivo or something similar to develop these descriptions based on the prominent use of the terms captured in their analysis? Please expand.

Results: My main concern with the use of angles to describe balance is that, as these are trunk and pelvic inclinations, they to not account for the conformation of the horse. I think it would be much more valuable data if you had calculated the inclination relative to their standing posture. So, if the hindlimbs were more engaged the angle would increase relative to the horizontal for example. What you are indicating is that a certain posture during the preparation phase is necessary to remain stable prior to performing the AAG. Your SD’s certainly show that the between horse data is variable. These results could be improved. If you do not have conformation, it would also be an option to use a comparison to an angle recorded prior to the preparation phase that is identifiable consistently and compare to that. This would show the difference between the preparation phase posture and before the preparation phase posture (possibly piaffe)?

Line 376: This sentence is badly worded. Bent is not really the best description. Please describe in relation to your sign convention.

4.6 Limitations

Line 546: remove ‘jumped’ and include performed the courbette or croupade, or something similar.

Comments on the Quality of English Language

Minor editing required.

Author Response

Dear Mr, Mrs,

Here are some answers,

“Line 232 to 233: ‘The rotation from an terrestrial reference frame to an inertial reference frame’. These are not terms that are generally used in biomechanics. Usually, the terms are global coordinate system or global reference frame and local reference frame or segment coordinate system are more commonly used.”

I also note that in a recent article (https://doi.org/10.3390/s23249625), the word “terrestrial” was used.

“In addition, having looked up the sensors you are using, they appear to be based on typical IMU technology, using gyros, accelerometers and magnetometers that have an internal reference frame relative to whatever the IMU is attached to. This then needs to be transformed if you are interested in global coordinates using quarternions to estimate the pelvic and sternum inclinations.”

“Is the appendix that you have provided from your own calculations? If so, can you please explain why you are using this method to measure local coordinates when the sensors, from the technical specification, are measuring in local coordinates? If this is from reference [21] this should not be in your appendix. It just needs to be more clearly explained here.”

Reference 21 describe how quaternion rotation is computed by Shimmer IMU sensor, from gyros, accelerometers and magnetometers. We did not implemented this computation, as their result is provided with sensors data. However, we quote reference 21 both in main text and in appendix, so man can implement quaternion computation for other IMU. We proposed a new formulation of the paragraph describing quaternions in appendix, hoping to be clearer.

Results: …

We wanted to use the global reference frame and not the horse reference during the piaffe or standing still at the beginning of the session. The main reason for this choice is that we wanted to quantify what the rider sees. But this is a calculation that can be made for comparison purposes.

Sincerely

Sophie Biau

Sophie Biau